# Unveiling the Effect of Solvents on Crystallization and Morphology of 2D Perovskite in Solvent-Assisted Method

**DOI:** 10.3390/molecules27061828

**Published:** 2022-03-11

**Authors:** Yingjie Su, Jianqiang Xue, Anmin Liu, Tingli Ma, Liguo Gao

**Affiliations:** 1State Key Laboratory of Fine Chemicals, Dalian University of Technology, Dalian 116023, China; suyingjie@mail.dlut.edu.cn (Y.S.); dut_jianqiang_xue@163.com (J.X.); anmin0127@163.com (A.L.); 2Department of Materials Science and Engineering, China Jiliang University, Hangzhou 310018, China; tinglima@dlut.edu.cn; 3Graduate School of Life Science and Systems Engineering, Kyushu Institute of Technology, Kitakyushu 808-0196, Fukuoka, Japan

**Keywords:** 2D perovskite, perovskite solar cells, crystallization orientation, solvent

## Abstract

Controlling the crystallographic orientations of 2D perovskite is regarded as an effective way to improve the efficiency of PSCs based on 2D perovskite. In this paper, five different assistant solvents were selected to unveil the effect of solvents on crystallization and morphology of 2D perovskite in a solvent-assisted method. Results demonstrated that the effect of Lewis basicity on the crystallization process was the most important factor for preparing 2D perovskite. The stability of the intermediate, reacted between the solvent and the Pb^2+^, determined the quality of 2D film. The stronger the Lewis basicity was, the more obvious the accurate control effect on the top-down crystallization process of 2D perovskite would be. This could enhance the crystallographic orientation of 2D perovskite. The effect of Lewis basicity played a more important role than other properties of the solvent, such as boiling point and polarity.

## 1. Introduction

Two-dimensional (2D) perovskite has been recognized as an alternative to its 3D analogs due to its excellent stability under ambient conditions [1,2]. However, perovskite solar cells (PSCs) based on 2D perovskite suffered lower power conversion efficiency (PCE) [3]. This is because of the introduction of bulky organic cations in 2D perovskite. On the one hand, the bulky organic cation leads to the formation of multi-quantum wells in 2D perovskite, which results in much larger exciton binding energy in 2D perovskite [4]. On the other hand, the insulating properties of the bulky organic cations will inhibit carrier transport in the out-of-plane direction in 2D perovskite [5]. Due to thermodynamic and kinetic reasons, 2D perovskites tend to grow along the in-plane orientation, especially lower-n phases, which makes the adverse effects of bulky organic cations more obvious [3].

Since the vertical orientation of 2D perovskite is important for PSCs, much attention has been paid to adjusting its crystal orientation to achieve highly oriented 2D perovskite thin films and photovoltaic performance [6,7]. According to the literature, 2D perovskites are crystallized in a top-down order: at the initial stage of crystallization, 3D-like perovskite is formed at the gas–liquid interface of the perovskite precursor due to the evaporation of the solvents; then the growth of lower-n phase perovskite is modeled by the 3D-like perovskite layer [8]. However, there will be nucleation inside the perovskite precursor during the crystallization process, which will lead to low crystallinity and random orientation. By adjusting the thermodynamics or dynamics of the perovskite crystallization process, the growth of perovskite can be controlled, such as slowing down the crystallization rate [9,10], strengthening the interaction of molecules between layers [11], and improving the temperature [3,12], pressure, or atmosphere during the crystallization process [7,13]. This is because the adjustment of the crystallization rate not only affects the orientation of perovskite, but also improves the efficiency of PSCs based on 2D perovskite. Therefore, a large part of the research on the adjustment of the 2D perovskite film formation process has been devoted to controlling its crystallization rate [14]. For example, additives (such as Cl^−^, SCN^−^) and solvents (such as dimethyl sulfoxide (DMSO), dimethylacetamide (DMAC)) were introduced into the precursor to combine with Pb^2+^ to form intermediates [15,16,17,18]. Additionally, the presence of intermediates could slow down the Pb^2+^ release rate during the crystallization process, resulting in decreasing the crystallization rate.

In our previous work, we proposed a solvent-assisted method to prepare 2D perovskite films, which could accurately control the crystallization order of 2D perovskite. The assistant solvent (DMSO) was distributed in a gradient in 2D perovskite [19]. However, the mechanism of the solvent-assisted method was not completely clear about which property of the assistant solvent is the main factor affecting the crystallization of 2D perovskite. Therefore, five different assistant solvents were selected and their effects in solvent-assisted 2D perovskite crystallization were compared. Results demonstrated that among various properties, the effect of Lewis basicity on the crystallization process was the most important factor for preparing 2D perovskite by the solvent-assisted method. This was because the solvent with strong Lewis basicity could form an intermediate with the Pb^2+^ with Lewis acidity, which could slow down the crystallization rate. It made the solvent-assisted method have a more obvious effect on the accurate control of 2D perovskite crystallization. Compared with Lewis basicity, the effect of the boiling point and polarity of the solvent was secondary. Therefore, in order to achieve the effect of slowing down the crystallization rate and controlling the crystallization more accurately, a solvent with stronger Lewis basicity should be selected.

## 2. Results and Discussion

Five commonly used solvents have been selected in the preparation of 2D perovskite, dimethylpropyleneurea (DMPU), DMSO, DMAC, N-methyl-2-pyrrolidone (NMP), and γ-butyrolactone (GBL). Their corresponding molecular formulas and properties are shown in Appendix A. The composition of 2D perovskite was (BA)_2_(MA)_3_Pb_4_(I_0.98_Cl_0.02_)_13_, which was prepared in air (25 °C, 45% relative humidity). The X-ray diffraction (XRD) results (Figure 1a) showed that two main peaks appeared at 14.28° and 28.58° for all 2D perovskite films, corresponding to the (111) and (202) plane [3]. The ratio of (111) and (202) peak intensity (I(111)/I(202)) has a relationship to the crystallographic orientations of 2D perovskite [20]. The smaller ratio meant that the 2D perovskite grew along the vertical orientation. By comparing I(111)/I(202), perovskite based on DMPU had the smallest ratio, where I(111)/I(202) was DMPU < DMSO < DMAC < NMP < GBL. It indicated that 2D perovskite based on DMPU solvent had better crystallographic orientation, and was more inclined to grow along the vertical orientation to the substrate. However, the ratio of 2D perovskite based on GBL solvent was the largest, indicating that the crystallographic orientation tended to grow parallel to the substrate surface. The full width at half maxima (FWHM) indicated that 2D perovskite based on DMPU had higher crystallinity, while the perovskite based on GBL had smaller crystal grains, as shown in Appendix A.

In order to explore the photophysical properties of 2D perovskite based on different solvents, UV visible (UV–Vis) absorption and photoluminescence (PL) spectroscopy tests were used, as shown in Figure 1b,c. There were four main absorption peaks at 567 nm, 598 nm, 634 nm, and 750 nm in the UV–Vis absorption spectrum, which represented the perovskite phase with *n* = 2, 3, 4 and ∞ [20]. This meant that 2D perovskites based on different solvents had the same structure. However, the perovskite based on DMPU had stronger light absorption intensity and higher PL intensity. It indicated that the 2D perovskite based on DMPU had better light absorption capacity, which could generate more carriers. The XRD results demonstrated that the 2D perovskite based on DMPU had better crystallization. It also meant that there was a smaller grain boundary and lower trap state density in 2D perovskite based on DMPU. Thus, the spontaneous radiative recombination was enhanced. The perovskite film based on GBL had the worst photophysical properties.

It could be seen that the perovskite films prepared by different assistant solvents had different qualities. The 2D perovskite based on DMPU had the best quality, followed by DMSO, DMAC, and NMP, while the worst was GBL. Combining the properties of five solvents, boiling point, polarity, and Lewis basicity, it could be clearly seen that the positive role that the assistant solvents played on 2D perovskite crystallization was consistent with the order of their Lewis basicity. It meant that the Lewis basicity property played the most important role in the preparation of 2D perovskite by the solvent-assisted method.

The architecture of PSCs was ITO/PEDOT: PSS/2D perovskite/PC_61_BM/BCP/Ag. The photovoltaic performance of devices based on different assistant solvents showed that the difference in short-circuit current (*J*_sc_) could be clearly observed in the current density–voltage (*J-V*) curve test (Figure 2a). In particular, the DMPU-based device showed the highest *J*_sc_, 19.94 mA/cm^2^, giving rise to a PCE of 13.69%. However, the *J*_sc_ of devices based on GBL was the smallest, which also obtained the lowest efficiency. The statistics of specific photoelectric parameters of devices based on different assistant solvents are shown in Table 1. Figure 2b shows the monochromatic incident photon-to-electron conversion efficiency (IPCE) curves and the integrated current curves of the five devices. They were consistent with the *J-V* test. The stabilized power outputs for the DMPU-based device had a steady-state current of 18.8 mA/cm^2^ at the maximum power point for 300 s, as shown in Figure 2c. The other steady-state output power curves are shown in Appendix A. All of them were consistent with the *J-V* test results, which proved that the reliability of the devices was good. In addition, we carried out *J-V* tests on 30 devices based on different solvents, and statistics of their optoelectronic parameters are in Table 1. In the histogram of the PCE statistics (Figure 2d), it can be observed that when DMPU was used as an assistant solvent to prepare 2D perovskite films, the distribution range of device efficiency was high and narrow. The enhancement of the photovoltaic performance and reproducibility of the devices based on DMPU were attributed to the improvement of the quality of the perovskite films, which had a strong correlation with the Lewis basicity of the assistant solvent.

To gain insight into the charge transfer dynamics of PSCs based on different assistant solvents, a series of characterization tests were carried out, including electronic impedance spectroscopy (EIS), the space charge limited current (SCLC), intensity-modulated photocurrent spectroscopy (IMPS), intensity-modulated photovoltage spectroscopy (IMVS), transient photocurrent (TPC), and transient photovoltage (TPV). The EIS test results were fitted according to the analog circuit diagram shown in the inset in Figure 3a. It could be seen that the DMPU-based device showed a larger semicircle, which meant that there was a large recombination resistance in the device [21]. Additionally, the GBL-based device showed the smallest semicircle. It indicated that the GBL-based device had the lowest recombination resistance. The order of the decreased recombination resistance of the devices was DMPU > DMSO > DMAC > NMP > GBL, as shown in Appendix A. It was consistent with the order of the assistant solvent’s Lewis basicity. To further explore the reasons, the trap state density of 2D perovskite films has been measured based on SCLC results. It was calculated by the dark *J-V* test (Figure 3b) on an electronic-only device (ITO/SnO_2_/2D perovskite/PC_61_BM/BCP/Ag). The calculated equation is:(1)$$N_{t}=\frac{2\epsilon_{0}\epsilon_{r}V_{\mathrm{TFL}}}{{\mathrm{qL}}^{2}},$$
where *N*_t_ is the trap state density, *V*_TFL_ is the trap-filling limit voltage, and *q* is the elemental charge [22]. The calculation results are listed in Appendix A. The trap state density of the DMPU-based device was 0.66 × 10^15^ cm^−3^, significantly lower than the other four devices, and the GBL-based device had the largest trap state density (2.10 × 10^15^ cm^−3^). This also explained why the former carrier recombination was significantly suppressed.

The IMPS and IMVS tests were used to study the carrier recombination loss in the devices based on different assistant solvents [18,23]. IMPS is often used to explore the bimolecular recombination of carriers in a device, as shown in Figure 3c. The relationship between *J*_sc_ and light intensity was shown as a power-law dependence, *J*_sc_ ∝ *I*^α^ (where I is the light intensity, α is the coefficient). α = 1 indicates that the carriers in the device have been completely transported away before recombination. The α value of the DMPU-based device was close to 1, indicating that the bimolecular recombination loss was the smallest. The IMVS test results showed the same regularity (Appendix A). When DMPU was used as the assistant solvent, the device exhibited the least loss of monomolecular recombination. The reduced carrier recombination in the device was consistent with the larger recombination resistance shown in EIS and the lower trap state density shown in SCLC. In order to further study the carrier dynamics, TPV and TPC were used to characterize the lifetime and extraction of carriers in the devices [21], which are shown in Appendix A and Figure 3d, respectively. The photovoltage decay time (τ_TPV_) and the photocurrent decay time (τ_TPC_) were calculated by quantizing a normalized transient photovoltage curve and normalized transient photovoltage curve with a single exponential decay model. The specific numerical statistics are shown in Appendix A. DMPU-based devices had higher τ_TPV_ and lower τ_TPC_, which meant that they had longer carrier lifetime and faster carrier extraction. This was considered to be the result of the lower trap state density of the photoactive layer. Through the characterization of the carrier behavior of the device, it could be seen that when DMPU was used as an assistant solvent, the perovskite film could have a lower defect state density. As a result, the carriers in the device had suppressed recombination, increased lifetime, and enhanced extraction and transport capabilities. Furthermore, the device showed a higher *J*_sc_ in the *J*-*V* test. More importantly, the order in which the transport capacity of the carriers in the device increases coincided with the order in which the assistant solvent Lewis basicity increases.

The stability test was carried out in air atmosphere (RH 50 ± 10%) without any encapsulation, and the results are shown in Figure 4a. The devices based on DMPU exhibited the best stability, which maintained 85% of the initial PCE after 1512 h. The DMSO- and DMAC-based devices maintained 76% and 71% of the initial PCE after 1512 h. However, the PCE of devices based on NMP and GBL declined rapidly, and dropped to 50% of the initial PCE after about 1512 h and 1360 h, respectively. The difference in device stability mainly came from the 2D perovskite layer. The DMPU-based 2D perovskite had a larger contact angle (Figure 4b). It meant its hydrophobicity could prevent the erosion of the perovskite by moisture, resulting in better stability.

The above tests illustrated that the order of 2D perovskite film quality and device performance was the same as the order of assistant solvent’s Lewis basicity. Compared with other properties of solvents, the Lewis basicity of the assistant solvent was the main factor affecting the perovskite film. It mainly affected the crystallization of 2D perovskite, as shown in Appendix A. It could be clearly seen that the crystallization rate of 2D perovskite was significantly reduced with the increase in Lewis basicity of assistant solvents. This was because Pb^2+^ (as Lewis acid) tended to combine with Lewis bases, where solvents with stronger Lewis basicity would compete with I^−^ for coordination sites around Pb^2+^. The stronger the Lewis basicity was, the stronger the binding ability to Pb^2+^. It caused a slower rate to release Pb^2+^ during the crystallization process. In our previous study, the assistant solvent had a gradient distribution in the vertical direction of 2D perovskite. The stronger the Lewis basicity was, the more obvious the accurate control effect on the top-down crystallization process of 2D perovskite would be. This could enhance the crystallographic orientation of 2D perovskite. The UV–Vis absorption spectra (Appendix A) showed that the stronger the Lewis basicity of the assistant solvent used, the later the absorption peak of the low *n* phase (*n* = 2) emerged. This was consistent with the above conclusion that the Lewis basicity of assistant solvent slowed down the crystallization rate, resulting in the low *n* phase perovskite appearing late. In order to improve the performance of 2D PSCs, an assistant solvent with strong Lewis basicity should be selected for solvent-assisted 2D perovskite crystallization.

## 3. Materials and Methods

### 3.1. Subsection

DMPU (99%), DMAC (for HPLC, ≥99.8%), NMP (99.9%), GBL (99.9%), and BCP (95%) were purchased from Aladdin. PbCl_2_ (99.999%), DMSO (hybridoma, 99.7%), N,N-dimethylformamide (DMF, for HPLC, 99.9%), chlorobenzene (for HPLC, 99.9%), and isopropanol (99.5%) were purchased from Sigma-Aldrich. PEDOT:PSS (Al 4083), BAI (99.5%), MAI (99.5%), and PC61BM (99.1%) were purchased from Xi’an Polymer Light Technology Corp. PbI_2_ (99.9%) was purchased from Advanced Election Technology CO,. Ltd. The 15 Ω/□ ITO was purchased from Yingkou OPV Tech New Energy Co.

### 3.2. Device Fabrication

The composition of BAI:MAI:PbCl_2_:PbI_2_ with a stoichiometric ratio of 2:3:0.08:3.92 was dissolved in DMF to prepare a perovskite precursor solution with a concentration of 0.8 M. Solvent-assisted method to prepare perovskite film: drop 160 μL solvent on the ITO with PEDOT:PSS, spin-coat at a low speed of 1000 rpm for 6 s, and dynamic spin-coat 60 μL the perovskite precursor solution at the 5th second, then spin at a high speed of 5000 rpm for 10 s, and then anneal at 100 °C for 15 min. Choose DMPU, DMSO, DMAC, NMP, and GBL as auxiliary solvents to prepare 5 different perovskite films. The specific deposition methods of the hole transport layer (PEDOT:PSS), electron transport layer (PC_61_BM/BCP), and silver electrode can be seen in the article we have published [19].

### 3.3. Characterization

The crystal structure and light absorption of 2D perovskite were measured using XRD (XRD-7000S, Shimadzu, Kyoto, Japan) and UV–Vis spectra (Lambda950, PerkinElmer, Buckinghamshire, UK), respectively. PL spectra were measured by a F-4500 (Hitachi, Tokyo, Japan) at an excitation wavelength of 495 nm. A Keithley 2460 (Keithley, Cleveland, OH, USA) was used to measure the *J*-*V* characteristics of cells at AM 1.5 G. IPCE was measured by using a computer-controlled xenon lamp combined with a monochromator (PEC-S20, Peccell, Kawasaki, Japan). EIS, TPC, TPV, IMPS, IMVS were all measured using an electrochemical workstation (Zennium Zahner, Kronach, Germany). In the IMVS tests, the relationship between *V*_oc_ and light intensity was shown on a logarithmic scale (linear relationship shown on a logarithmic scale), *V*_oc_ ∝ *S*ln(*I*) (where *I* is the light intensity, *S* = (*nkT*)/q, k is the Boltzmann constant, *T* is the room temperature in Kelvin, and q is the elementary charge). As *n* ≈ 1, the bimolecular recombination is dominant in devices and as *n* ≈ 2, the monomolecular recombination is dominant in devices.

## 4. Conclusions

In conclusion, five different solvents have been selected to unveil the effect of solvents on crystallization and morphology of 2D perovskite in a solvent-assisted method. Results demonstrated that the property of Lewis basicity played a more important role in the crystallization process for preparing 2D perovskite, compared with other properties of assistant solvents, such as polarity and boiling point. The quality of 2D perovskite film and the photovoltaic performance of devices was consistent with the order of Lewis basicity of solvents. The crystallization rate of 2D perovskite was significantly reduced with the increase in Lewis basicity of assistant solvents. By controlling the crystallographic orientations of 2D perovskite, 2D PSCs based on DMPU showed the best PCE and stability.

## Figures and Tables

**Figure 1 molecules-27-01828-f001:**
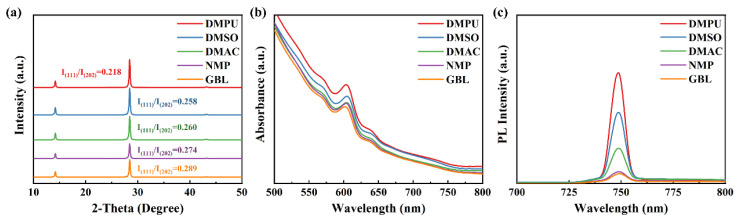
(**a**) XRD patterns, (**b**) UV–Vis absorption, and (**c**) PL spectra of the perovskite films prepared with different solvents.

**Figure 2 molecules-27-01828-f002:**
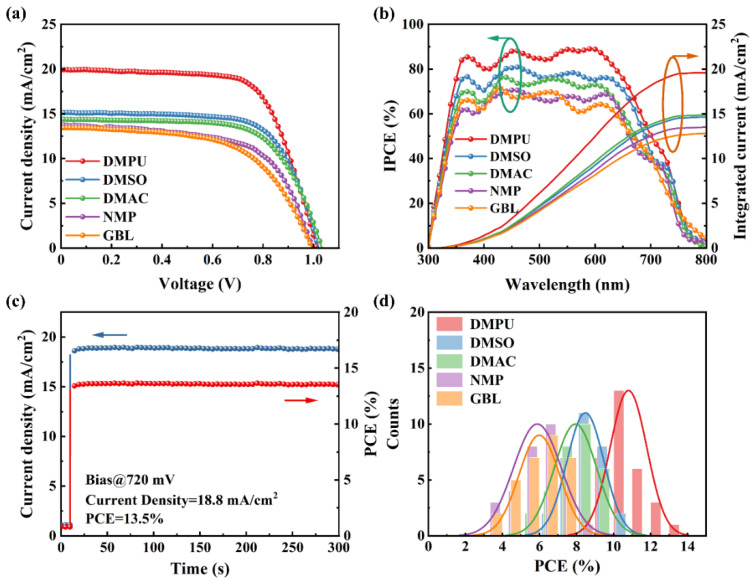
(**a**) *J-V* curves and (**b**) IPCE of devices prepared with different solvents. (**c**) the stabilized photocurrent measurement and power output of devices prepared by DMPU, under simulated AM 1.5 G illumination of 100 mW/cm^2^. (**d**) Histogram of PCEs of devices prepared with different solvents.

**Figure 3 molecules-27-01828-f003:**
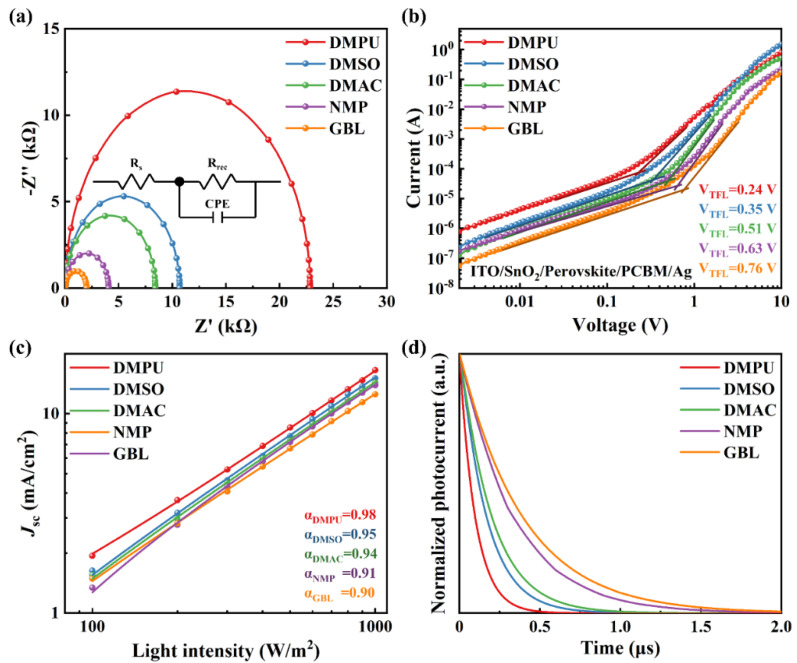
(**a**) EIS measurements of devices prepared with different assistant solvents. (**b**) Dark *J-V* characteristics of electron-only devices. (**c**) IMPS and (**d**) normalized TPC curves of devices prepared with different solvents.

**Figure 4 molecules-27-01828-f004:**
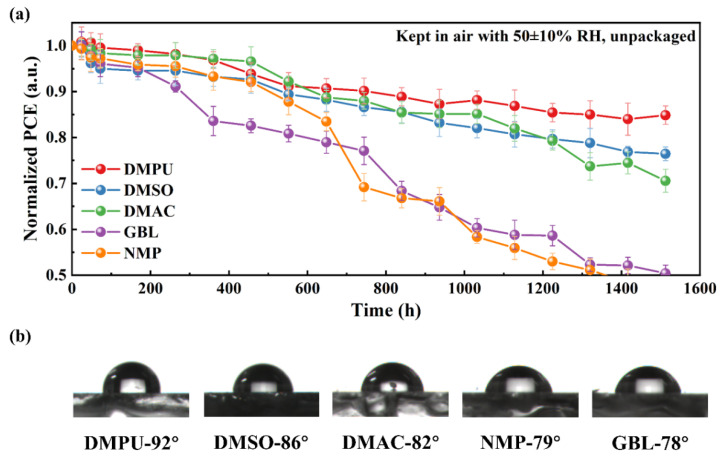
(**a**) PCE decay measurements based on unpackaged devices prepared by different solvents in air (average values were obtained based on 3 devices in each condition). (**b**) Snapshots of surface contact angle measurements for perovskite films prepared with different solvents.

**Table 1 molecules-27-01828-t001:** Photovoltaic parameters of solar cells based on perovskite films prepared with different solvents.

		*V*_oc_ (V)	*J*_sc_ (mA/cm^2^)	FF	PCE (%)
DMPU	best	1.01	19.94	0.68	13.69
	average	1.00 ± 0.06	17.02 ± 0.98	0.67 ± 0.08	12.19 ± 0.93
DMAC	Best	1.01	15.14	0.69	10.58
	average	0.97 ± 0.06	14.18 ± 1.83	0.68 ± 0.07	8.48 ± 0.98
DMAC	best	1.02	14.94	0.65	9.88
	average	0.97 ± 0.06	13.78 ± 0.96	0.63 ± 0.08	7.94 ± 1.12
NMP	best	0.99	13.75	0.63	8.49
	average	0.97 ± 0.09	12.02 ± 1.02	0.63 ± 0.05	5.89 ± 1.27
GBL	best	0.99	13.44	0.59	7.80
	average	0.98 ± 0.07	10.46 ± 1.7	0.59 ± 0.05	6.0 ± 1.08

There are 30 devices taken into account to calculate the average parameters of devices.

## Data Availability

Data will be available from the corresponding author upon logical request.

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
