# Peer review of "Unveiling the Effect of Solvents on Crystallization and Morphology of 2D Perovskite in Solvent-Assisted Method"

_molecules, 2022, doi:10.3390/molecules27061828_

Round 1

Reviewer 1 Report

The manuscript concentrated on influence of five solvents (DMPU, DMSO, DMAC, NMP, GBL) on crystallization and morphology of 2D perovskite in solvent-assisted method. Results demonstrated that among various properties, the effect of Lewis basicity on the crystallization process was the most important factor for preparing 2D perovskite by solvent-assisted method. Moreover, the authors found that the 2D PSCs based on DMPU was the best.

The manuscript was written well. In my opinion this article is worth to be published in molecules after minor revision. I have just few comments, which authors may consider prior to publication of this work:

  1. The first sentence in the abstract “Two-dimensional (2D) perovskite performed as an alternative to their 3D analogs due to their excellent stability, but perovskite solar cells (PSCs) based on 2D perovskite suffered lower power conversion efficiency (PCE) due to the multi-quantum-wells in 2D perovskite and insulating properties of the bulky organic cation.” can be moved to the introduction not suitable in the abstract. Moreover, the findings of the PSCs should be reflected in the abstract.

  1. I encourage the authors to calculate the crystallite size of the 2D perovskite prepared with the different five solvents.
  2. In lines 161 & 162 replace ×1015 by ×1015.
  3. Add the references at end of this sentence: “The specific deposition methods of the hole transport layer (PEDOT:PSS), electron transport layer (PC61BM/BCP) and silver electrode can be seen in the articles we have published. “
  4. Check both Funding and Acknowledgments. They look as it in the Template!

Author Response

Reponse to Reviewer #1:

  1. The first sentence in the abstract “Two-dimensional (2D) perovskite performed as an alternative to their 3D analogs due to their excellent stability, but perovskite solar cells (PSCs) based on 2D perovskite suffered lower power conversion efficiency (PCE) due to the multi-quantum-wells in 2D perovskite and insulating properties of the bulky organic cation.” can be moved to the introduction not suitable in the abstract. Moreover, the findings of the PSCs should be reflected in the abstract.”

Answer: Thanks. The first sentence has been removed from introduction. And the corresponding findings have been added in the abstract, which is marked as cyan.

  1. I encourage the authors to calculate the crystallite size of the 2D perovskite prepared with the different five solvents.”

Answer: Thanks. The crystallite size of the 2D perovskite prepared with the different five solvents has been calculated, and the corresponding results were shown in Figure S1, which was marked as green.

  1. In lines 161 & 162 replace ×1015 by ×1015.”

Answer: Thanks. We have corrected this formatting error and marked it in red.

  1. “Add the references at end of this sentence: “The specific deposition methods of the hole transport layer (PEDOT:PSS), electron transport layer (PC61BM/BCP) and silver electrode can be seen in the articles we have published.”

Answer: Thanks. We have added reference at end of the above sentence and marked it as blue typeface.

  1. Check both Funding and Acknowledgments. They look as it in the Template!”

Answer: Thank you so much. We have added the contents of Funding and Acknowledgments, which were marked as gray.

Reviewer 2 Report

The manuscript entitled “Unveiling the Effect of Solvents on Crystallization and Morphology of 2D Perovskite in Solvent-Assisted Method” is original and interesting.

However, the reviewer suggests minor revision:

  • In some sentences, English should be improved e.g.

(Line 38-41) “Since the vertical orientation of 2D perovskite is important for PSCs based on 2D perovskite, much effort has been paid to adjust crystallographic orientations of 2D perovskite in order to achieve highly oriented 2D perovskite thin films and photovoltaic performance[6,7].” - remove the repetitions of words.

(Line 13-16) “Two-dimensional (2D) perovskite performed as an alternative to their 3D analogs due to their excellent stability, but perovskite solar cells (PSCs) based on 2D perovskite suffered lower  power conversion efficiency (PCE) due to the multi-quantum-wells in 2D perovskite and insulating properties of the bulky organic cation.” - remove the repetitions of words.

  • Line 261-263 “Results demonstrated that the property of Lewis basicity played more important role on the crystallization process for preparing 2D perovskite.” - The authors should explain – comparison to what - “….more important role than………….”

Author Response

Response to Reviewer #2:

Comments to the Author: “The manuscript entitled “Unveiling the Effect of Solvents on Crystallization and Morphology of 2D Perovskite in Solvent-Assisted Method” is original and interesting.

However, the reviewer suggests minor revision:

In some sentences, English should be improved e.g.

(Line 38-41) “Since the vertical orientation of 2D perovskite is important for PSCs based on 2D perovskite, much effort has been paid to adjust crystallographic orientations of 2D perovskite in order to achieve highly oriented 2D perovskite thin films and photovoltaic performance[6,7].” - remove the repetitions of words.

(Line 13-16) “Two-dimensional (2D) perovskite performed as an alternative to their 3D analogs due to their excellent stability, but perovskite solar cells (PSCs) based on 2D perovskite suffered lower  power conversion efficiency (PCE) due to the multi-quantum-wells in 2D perovskite and insulating properties of the bulky organic cation.” - remove the repetitions of words.

Line 261-263 “Results demonstrated that the property of Lewis basicity played more important role on the crystallization process for preparing 2D perovskite.” - The authors should explain – comparison to what - “….more important role than………….”

Answer: Thanks. We have revised the above sentences and marked them as yellow. In addition, we also carefully check and revise other sentences in the whole paper.
